# A Framework for Resource Allocation in Fire Departments: A Structured Literature Review

**Milad K. Eslamzadeh** *, **António Grilo** and **Pedro Espadinha-Cruz**

UNIDEMI, Department of Mechanical and Industrial Engineering, NOVA School of Science and Technology, Universidade Nova de Lisboa, 2829-516 Almada, Portugal; acbg@fct.unl.pt (A.G.); p.espadinha@fct.unl.pt (P.E.-C.)
* Correspondence: s.eslamzade@campus.fct.unl.pt

**Abstract:** A significant amount of research has been conducted on the resource allocation in fire departments (RAFD) and literature reviews about the fire protection service (FPS), but to the best of our knowledge, no literature reviews have been conducted about the RAFD. Therefore, the purpose of this research is to review literature about allocating resources to urban fire departments (FDs) to gain state-of-the-art knowledge of RAFD and identify the most frequent methodologies and measures in the studies. A five-stage structured literature review (SLR) is undertaken to analyze the RAFD-related studies; subsequently, statistical analysis is used to disclose additional information from the retrieved data and develop a general framework for RAFD. According to the structured literature review, which yielded 417 independent variables for RAFD, integer programming (IP) and data envelopment analysis (DEA) are the most common approaches for RAFD among the mathematical and statistical models in the evaluated articles. Based on the findings, a general conceptual framework for RAFD is suggested. The findings of this study can help public and private FDs and FPS managers, decision-makers, resource allocation (RA) researchers, and academicians gain state-of-the-art knowledge of RAFD. The proposed RAFD framework can provide the FPS decision-makers with the appropriate method and variables for allocating their limited resources in a more efficient way within their FDs.

**Keywords:** resource allocation; performance assessment; fire risk assessment; fire protection services; fire departments; structured literature review

## 1. Introduction

One of the most critical challenges in urban management is providing efficient and effective fire protection services (FPS) as a vital part of human and environmental safety. Along the same line, the FPS's efficiency and effectiveness are highly dependent on the efficient resource allocation in the fire departments (RAFD) and locating the fire stations [1,2].

The RAFD is among the main concerns of the FPS decision-makers [3]. It is a strategic decision in the public sector management that differs from the traditional private resource management. The FPS's objectives include minimizing the number of fire incidents and maximizing the saved lives and properties (i.e., social cost minimization), but for the private sector, they include maximizing profit or minimizing cost [2,4].

A significant concern of FPS decision-makers for RAFD is urban expansion and resource constraints [5,6]; therefore, allocating the limited resources among fire departments (FDs) to provide FPS to more people in less time is a goal that is highly dependent on selecting an appropriate RAFD method and variables [7,8]. Although many studies have proposed different methods and applied a wide variety of variables to satisfy that goal, not all were adequate for RAFD [9–12]. Consequently, an exhaustive review of the RAFD studies is required to extract the RAFD methods and relevant variables from the literature and identify the most frequent ones. These results can help to construct a general framework for RAFD that can provide the FPS decision-makers with the appropriate method and variables for allocating their limited resources in a more efficient way within their FDs.

In spite of numerous literature reviews on the FPS's coverage problem [13–15], station location problem [16,17], facilities location problem [18–20], and fire risk assessment [21–23], to the best of our knowledge, no reviews have been carried out about RAFD. Hence, the objective of this study is to provide state-of-the-art knowledge on RAFD and to identify the most frequent RAFD methods and variables by synthesizing the results of the structured literature review (SLR). Another contribution of this research includes providing the general RAFD framework that could be used for the forthcoming RAFD both in public and private FDs.

The remainder of the paper is divided into five sections. The research methodology is described in Section 2, and then the SLR's findings and outcomes are presented and discussed in Section 3. Section 4 introduces and explains the general RAFD framework and eventually, this paper concluded in Section 5.

## 2. Research Methodology

To provide state-of-the-art knowledge of RAFD, an SLR is used in this paper and its findings are used to design and develop a general framework for RAFD. An SLR is "a method for studying a corpus of scholarly literature, to develop insights, critical reflections, future research paths, and research questions" [24], and from a process-based perspective, "An SLR offers an empirical grounding that avoids missing seminal articles and reduces researcher bias" [25].

Depending on the SLR's objectives and questions, four to ten SLR stages were mentioned in different studies [24,26–28]. Following the previous studies [2,28,29], a five-staged SLR is used in this research, which is as follows:

1. *Defining Stage*: defining the problem and research questions.
2. *Searching Stage*: searching scientific repositories.
3. *Screening Stage*: screening the results (using RAYYAN platform [30]).
4. *Coding Stage*: coding the selected literature (using MAXQDA software [31]).
5. *Synthesis Stage*: aggregating and presenting the results.

Figure 1 illustrates the SLR process flow of this study.

### 2.1. SLR Protocol Creation Stage

RAFD is the main focus of many studies, and where they applied different methods and variables to calculate it. In spite of the review studies that have been carried out in the FPS domain, as far as our knowledge, no work has been conducted to investigate the RAFD literature and synthesis for the most common and frequent methods and variables. Hence, an SLR was conducted to find out the following:

- RQ1—What is the state of the art of the RAFD?
- RQ2—Which are the most applied RAFD methods in the studies?
- RQ3—What are the most employed variables for the RAFD?

### 2.2. Search Stage

This process of searching for the RAFD papers in the search stage started with selecting the scientific repositories and a Boolean query [2,24,27]. Table 1 shows the search phrases, science databases, and inclusion-exclusion criteria that were used in the search stage of the SLR, which resulted in identifying 5329 papers, as shown in Figure 1.

The search query covers the two parts of the research, which are "resource allocation" and its relevant keywords (the first parenthetical phrase) and "fire departments" and its similar keywords (the second parenthetical phrase). To guarantee that the search results were the most relevant studies to RAFD and contained all of the combinations of the mentioned keywords, the wildcard command or "*" was used at the end of the keywords, and the logical "AND" joined the phrases.

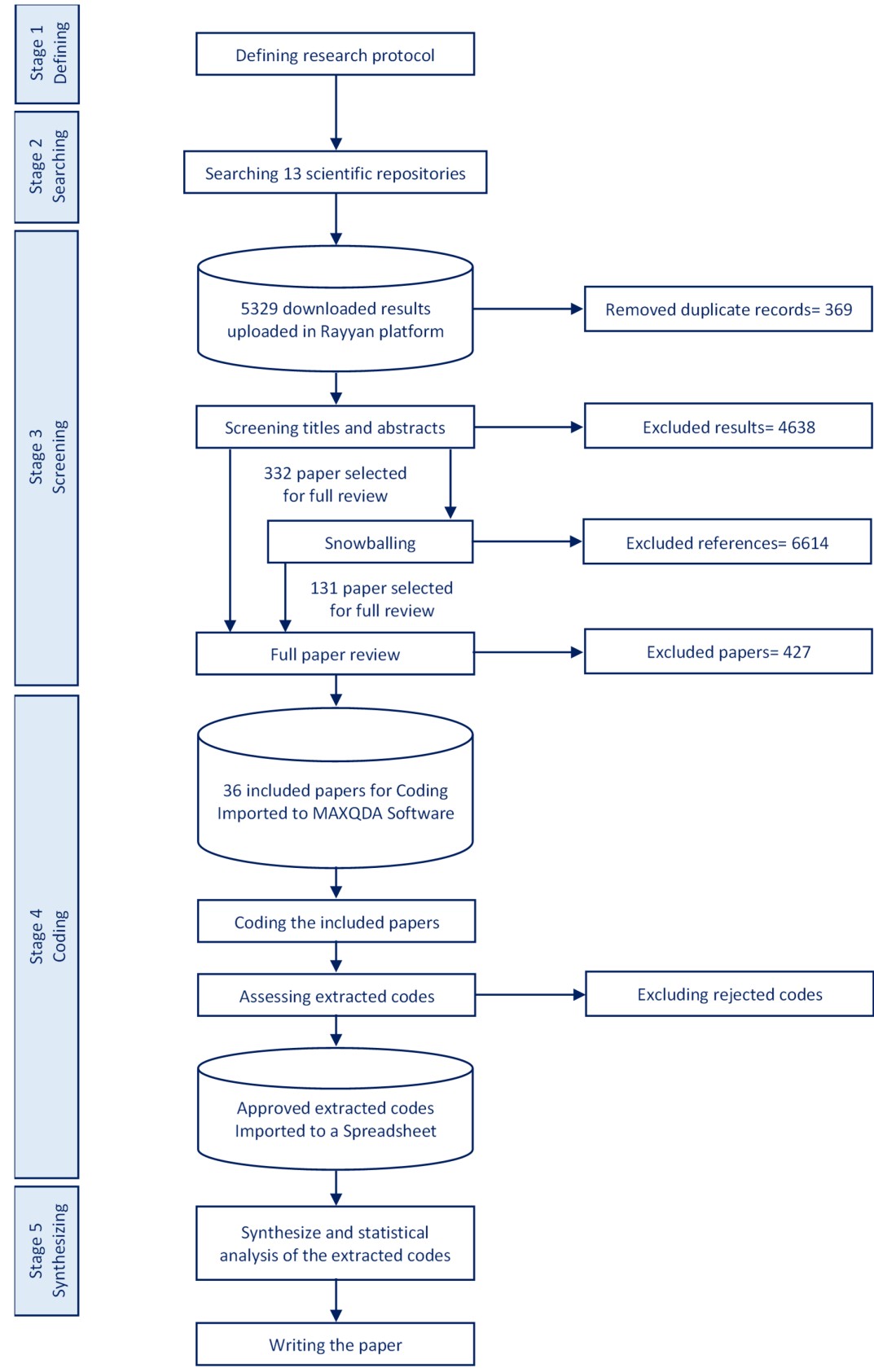

**Figure 1.** The SLR process flow of this research.

**Table 1.** The SLR's search phrase, science databases and inclusion-exclusion criteria.

| | |
|---|---|
| **Boolean Query.** | **(("Resource Alloc*" OR "Resource Manag*" OR "Resource Plan*" OR "Resource Sharing") AND ("Fire Station*" OR "Fire Department*" OR "Urban Fire" OR "Fire Incident*" OR "Emergency"))** |
| Scientific repositories | Science Direct (Elsevier, Scopus), IEEE Xplore, JStor, Web of Science, Wiley Online Library, Emerald, Taylor and Francis, Springer, Sage Online, EBSCO, Oxford Academic, ESO (European Source Online), and ScienceOpen. |
| Inclusion criteria | - English language papers<br>- Peer-reviewed papers<br>- Strategic level RAFD<br>- Urban-residential area<br>- The publication period did not have a limit (to have all the possible results)<br>- Articles about resource allocation in both private and public FDs were included (to investigate the differentiation and similarities) |
| Exclusion criteria | - Book chapters<br>- Reports<br>- Non-English language papers<br>- Operational level RAFD<br>- Irrelevant resource allocation studies |

*2.3. Screening Stage*

For screening the titles and abstracts of the identified results, all the results from the searching stage were uploaded to the "Rayyan" online platform [30].

The following inclusion criteria were used during the screening step to find the most relevant RAFD studies for this investigation, as shown in Table 1, and in accordance with the following criteria proposed by [2]:

- *Organizational level*: the investigated studies fall into one of two significant categories by their evaluations, which are as follows:

  ○ *Operational:* These are the studies about resource allocation (RA) at the fire scene at an operational level, not the fire department. This type of research considers the specifications and behavior of fire, such as the number of firefighters in the operational ream, unit size for onsite scenarios, vehicle dispatch models, and vehicle routing. We excluded them from our review.

  ○ *Strategic:* These are studies about RA in the fire department that are more managerial in nature. For instance, these papers evaluate the performance of the FDs and stations, fire risk assessment, and demand prediction to do the RAFD. These publications were taken into account for the remainder of the investigation.

- *Urban-Residential areas*: Since the forest and wildfires differ from urban-residential fires in terms of hazard type, losses, accessibility, and suppression requirements, this research was focused on those RAFD studies that considered the characteristics of the urban-residential areas in their model.

- *Fire department's resource allocation:* Many studies have been conducted about RA in different domains, but the specifications of the fire departments and the type and usage of the resources vary in their nature and application. Therefore, some research subjects were excluded from this study, making sure only the most relevant research was investigated in detail. During the screening stage, the following RA subjects were excluded because they either fell into the operational level category or were not relevant to urban-residential area fire services:

  ○ FPS unit dispatch, fire vehicles routing.
  ○ Emergency medical services, hospital resources, staffs, and medical emergency vehicles (e.g., ambulances).
  ○ Pre/post-disaster relief supplies: location allocation and RA related to the earthquake, flood, natural disaster and terrorist attack and relief supplies storage.

○ Resource conflict checking and resolution control.
○ Location and coverage problems that solely optimize the station locations, spare part inventory, context-free models, server locations.
○ RA models for road and marine accidents, police and patrolling, aerial supplies, public-private transportation, costal and in-road supplies.

Figure 1 illustrates the process of the SLR stages, including the screening stage. The first step in this stage was removing 369 duplicated results. Then, by considering the inclusion criteria, all the titles and abstracts of the results were investigated and in this stage, 4638 results were excluded and 332 papers were selected for full paper reviews. Checking the references and citations of the selected papers, which is called snowballing or chain sampling [32], constituted the next step and during the snowballing step, 6614 references and citations were checked and 131 of them were also selected for a full review. By fully reviewing the 463 selected papers, 36 papers were included in the coding stage.

*2.4. Coding Stage*

2.4.1. Validity

According to [2,24], the validity check was carried out for this research to make sure that important evidences were taken into consideration and were not avoided in order to find an easy solution. Therefore, besides applying the inclusion criteria in the searching and screening stages (i.e., including only articles from peer-review journals to "guarantee a quality process in selecting meaningful contributions" and "the best available literature on the subject filling the gap between theory and practice" [33]), the pattern-matching approach [24] was also applied. The coding criteria and list of code titles were developed in this step by studying a set of five articles and confirming them with an expert's view, and then the other papers were coded [2,33]. The expert committee consisted of the chief commander of the Portuguese National Authority for Emergency and Civil Protection (ANEPC), the dean of the national firefighting academy, a full professor, and an assistant professor with pertinent qualifications and studies in FPS management.

The authors re-evaluated and confirmed all of the extracted codes; also, along the same line as [34], it was established via the authors' discussions with the strategic management of the ANEPC that the findings are consistent with their experience.

2.4.2. Reliability

All of the results were double-checked by the two authors during the coding step, just as they were by [2,33]. It follows Hayes and Krippendorff's recommendation (2007) that "it should assess the agreement between two or more observers who describe each of the units of analysis separately from each other"; thus, this study could use Krippendorff's alpha inter-coder reliability test to check its coding reliability. Only those codes with both authors' agreement were included in the quality assessment procedure (please see Figure 1); hence, Krippendorff's alpha for all the included codes was 1.000 [34].

2.4.3. Coding Process

The contents of the selected articles were examined at this stage. This entailed taking qualitative and quantitative data and codifying the data into a pre-defined set of codes in order to find trends in RAFD articles and present them in a systematic, objective, and trustworthy manner [35].

A database of coded data was generated after examining the publications and extracting the data from them. The coding step was finished by having rounds of dialogues with FPS experts to assess the quality of the codes. The coding was performed with the MAXQDA 2020 program [31], and the outcomes of the coding step are summarized in Table 2.

**Table 2.** Summary of the extracted codes in the coding stage.

| Ref | Data Acquisition | | | | Data Collection | | | Demand Prediction | Equity Method | Dynamic | Highly Vulnerable Area | Allocation Level | Socio-Economic | Spatio-Temporal | Case Study Country | FRA Model | PA Model | Coverage Model |
|---|---|---|---|---|---|---|---|---|---|---|---|---|---|---|---|---|---|---|
| | Unpublished Database | GIS | Published Reports | Sample/Generated Dataset | Survey | Observation-Experiment | Expert Opinion-Interview | | | | | | | | | | | |
| [36] | X | | X | | | | X | | X | | X | Station | | | Greece | Historical incident data | | |
| [37] | X | | | | | | | | X | | | DP | X | | UK | Historical incident data | DEA | |
| [38] | X | | | | | | | | | | | Station | | | Taiwan | Predicting the incidents (machine learning) | DEA | |
| [39] | | | X | | | | | | | | | DP | | | South Korea | - | DEA | |
| [40] | X | | | | | | | | | | | Station | | | Taiwan | Predicting the number of incidents (machine learning) | DEA | |
| [41] | X | | | | | | | | | | | DP | X | | Taiwan | - | DEA | |
| [42] | X | | | | | | | | | | | DP | X | | China | - | DEA | |
| [7] | X | X | X | | X | | X | | X | | X | Station | X | X | India | Predicting the number of incidents (machine learning) | | TIMEXCLP |
| [43] | X | | X | | | X | | | | | | Squad | | X | US | Historical number of incidents | DEA | |
| [5] | X | | X | | | | X | | | | X | State | X | | US | Fire risk score | DEA | |
| [10] | X | | | | | X | | | X | | X | Station | | X | Singapore | HVA locations | | IP |
| [44] | X | | X | | | X | | | X | | X | Station | | X | USA | - | | IP |
| [45] | X | | | | | | | | | X | X | Station | | X | Singapore | Generated/sample incidents dataset | | Probabilistic FAST |

**Table 2.** *Cont.*

| Ref | Data Acquisition | | | | Data Collection | | | Demand Prediction | Equity Method | Dynamic | Highly Vulnerable Area | Allocation Level | Socio-Economic | Spatio-Temporal | Case Study Country | FRA Model | PA Model | Coverage Model |
|---|---|---|---|---|---|---|---|---|---|---|---|---|---|---|---|---|---|---|
| | Unpublished Database | GIS | Published Reports | Sample/Generated Dataset | Survey | Observation-Experiment | Expert Opinion-Interview | | | | | | | | | | | |
| [46] | X | | X | | | X | X | | | X | X | Station | X | X | UK | Fire risk score | | Coverage model (no details provided) |
| [47] | X | X | | | | X | | | | | X | Station | X | X | USA | HVA locations | | Coverage model (no details provided) |
| [48] | | X | X | | | X | | | | X | X | Station | | X | Pakistan | Generated/sample incidents dataset | | Coverage model (no details provided) |
| [49] | X | X | X | | X | | X | | | X | X | Station | X | X | India | Predicting the number of incidents (machine learning) | | TIMEXCLP |
| [6] | X | | X | | | | X | | | X | X | Station | | X | China | Predicting the number of incidents (machine learning) | | MCLP-P |
| [11] | X | X | X | | | X | X | | | X | X | DP | X | X | Chile | Historical incident data | | HQM and FLEET-EXC |
| [50] | X | | X | | | | X | | | X | | Station | | X | Chile | Historical incident data | | IP |
| [8] | | | X | | | | | | | | X | Station | X | X | US | Generated/sample incidents dataset | | TEAM, MOTEAM, and FLEET |
| [51] | | | | X | | | | | | | | Station | X | X | Sample data | Generated/sample incidents dataset Fire risk score | | CMMSR |
| [52] | | | | X | | | | | | | | Station | X | X | Sample data | Generated/sample incidents dataset | | FAST, MCLP, and FLEET |

**Table 2.** *Cont.*

| Ref | Data Acquisition | | | | Data Collection | | | Demand Prediction | Equity Method | Dynamic | Highly Vulnerable Area | Allocation Level | Socio-Economic | Spatio-Temporal | Case Study Country | FRA Model | PA Model | Coverage Model |
|---|---|---|---|---|---|---|---|---|---|---|---|---|---|---|---|---|---|---|
| | Unpublished Database | GIS | Published Reports | Sample/Generated Dataset | Survey | Observation-Experiment | Expert Opinion-Interview | | | | | | | | | | | |
| [1] | X | | X | X | X | X | X | | | | X | Station | X | X | Belgium | Predicting the number of incidents (machine learning) | | MCLP, MFFNN and HQM |
| [53] | | | X | | X | | | | | X | | Station | | X | China | Historical incident data | | PDQC-M |
| [54] | | | | X | X | | | | | X | X | Station | X | X | | Generated/sample incidents dataset | | FMCVLM |
| [4] | X | | X | | X | X | | | | X | | Station | | X | Chile | Predicting the number of incidents (machine learning) | | MIPFMM |
| [12] | X | | | | X | X | | | | X | | Station | X | X | Chile | Predicting the number of incidents (machine learning) | | Robust FLEET-EXC |
| [55] | | | | X | | | | | | | | Station | | X | Sample data | Generated/sample incidents dataset | | MEMCOLA |
| [3] | X | | X | | X | | | | | | X | Station | | | Iran | Historical incident data Fire risk score | | IP |
| [56] | X | X | | | | | | | | X | X | Station | | X | China | HVA locations | | DEM |
| [57] | X | | X | X | X | X | | | | X | X | Station | X | X | Ukraine | Historical incident data | | IP |

**Table 2.** *Cont.*

| Ref | Data Acquisition | | | | Data Collection | | | Demand Prediction | Equity Method | Dynamic | Highly Vulnerable Area | Allocation Level | Socio-Economic | Spatio-Temporal | Case Study Country | FRA Model | PA Model | Coverage Model |
|---|---|---|---|---|---|---|---|---|---|---|---|---|---|---|---|---|---|---|
| | Unpublished Database | GIS | Published Reports | Sample/Generated Dataset | Survey | Observation-Experiment | Expert Opinion-Interview | | | | | | | | | | | |
| [58] | X | | | | | | | | | | | Station | | X | Iran | Predicting the number of incidents (simulation) | | Coverage model (no details provided) |
| [9] | | | | X | | | | | | X | X | Station | | X | Sample data | Generated/sample incidents dataset | | FDMCLAP, PSO, and ABC |
| [59] | X | | | | | X | | | | X | | Station | | X | Canada | Predicting the number of incidents (machine learning) | | Coverage model (no details provided) |
| [60] | X | X | | | | X | | | | X | X | Station | | X | China | HVA locations | | FARS |

*2.5. Synthesis Stage*

The statistical analysis of codes in each criterion was completed at this stage. The findings of the statistical analysis are presented and discussed in Section 3 in the form of charts and tables, and Section 4 introduces a general RAFD framework based on the supplied results.

**3. Findings and Discussion**

The synthesis stage's conclusions are based on a statistical examination of Table 2's data. This part includes the findings, as well as the pertinent debates.

*3.1. Research Identifications*

The 36 studied papers were published in 30 different journals between 1979 and 2021, indicating that there has been a continuous priority over improving FPS coverage and performance through effective RAFD over the past four decades. This domain has piqued the interest of many scientific journals, due to its importance and impact on public welfare and safety. Because of the snowball approach in the SLR search stage, which ensured the comprehensiveness of the search process by screening all relevant works from the same researchers [35], some authors had more than one publication in the coded RAFD papers.

*3.2. Data Specifications*

3.2.1. Data Sources

Data is crucial in resource allocation (RA) research, and finding reliable and high-quality data is a difficulty for every study. Data collection and data acquisition were the two primary data gathering sources among the coded publications. Researchers and RAFD assessors can use this information to learn more about the key data sources used in RAFD studies and to gain a better grasp of how to obtain the data they want for their investigations.

As shown in Table 2, data acquisition is the most popular method for data collection among the reviewed studies; 27 of the 36 investigated articles had access to unpublished databases to acquire their data, while 17 relied on the data in publicly available reports. Six of the studies also used open-access geographic information system (GIS) data, and seven of them used sample or generated datasets in place of actual data.

In terms of data collection, expert opinion was employed in almost one-third of the studied RAFD literature (13 studies), survey questionnaires were used in four papers, and just one paper used site visits and first-hand observation. In 15 studies, a mix of collecting and acquiring strategies from various sources was applied.

3.2.2. Dynamic vs. Static Data

The output and outcome of the FPS are highly dependent on the timely services and time of arrival of firefighting units at the fire scene, which has a critical role in casualty reduction [2]. There are various dynamic aspects that have a significant impact on the travel time of firefighting units (e.g., traffic, time of day, vehicle types and speeds, seasonal fluctuations) that should be included in the RAFD evaluation [7], and 19 papers used dynamic data. Despite the importance of dynamic data, nearly half of the studies ignored it and relied solely on static data (e.g., number and location of events, and distance from stations) or the average values of dynamic variables (17 papers out of 36, to be precise). The researchers collected data on traffic, travel duration, speed, and daily or seasonal variations in congestion patterns using online maps and GIS, as well as FPS authorities' internal data.

3.2.3. Highly Vulnerable Areas (HVA) Data

There are infrastructures, buildings, and neighborhoods in the urban and residential areas that are especially vulnerable to fire occurrences, owing to their use, residents, or activities, and the fire casualties in these areas are quite high. Hospitals, schools, energy and transportation infrastructures, and high-rise buildings, among other extremely sensitive

areas, require particular fire protection and prompt reactions in the event of a fire [44,49]. As a result, decision-makers and RAFD assessors must take into account these sorts of structures in the RAFD process. In the reviewed paper, the authors considered the HVA by adding them as weights in their RAFD models. Some examples of HVA variables are *building usage* (residential, commercial, industrial, and mixed categories) and *built-up compactness parameter* (average floor and heights in the area) [7], *vulnerability level* and *fire risk score* (least vulnerable-1 to most vulnerable-6) [5], *the environmental risks* (a combination of 50 variables, such as facility access to water, terrain type, and construction type) [1], and *number of high-risk sites* [57].The publicly available data, online maps and GIS, and FDs' internal data are the sources for acquiring the HVA data. As reported in Table 2, surprisingly, 16 papers out of 36 studies did not include the HVA in their RAFD models.

To summarize this section, the author might note that, in addition to the FDs' private information, publicly accessible and online data (e.g., population size, density, and traffic data) should also be taken into account while obtaining the data for RAFD assessments. Another critical element of the RAFD evaluations is utilizing the HVA data.

### 3.3. Research Characteristics and Assessment Methods

RAFD studies differ in their approach to RA and how they consider the importance of each incident, demand, station, or FD in their RAFD models to cover the demands. In this section, the most common approaches, models and their categories in the investigated studies are introduced and discussed in detail.

#### 3.3.1. RAFD Approaches

According to Table 2, all of the examined articles employed one or more of the following three basic approaches to RAFD that are described in detail in the following sections:

- *Coverage problem (CP)*: Based on demand points and potential facility sites, emergency facility location problems have traditionally included considerations regarding which places should be selected as facility depots and how many facilities should be placed in each depot. A variety of models have been created to tackle facility location concerns [15]. The RAFD process in this category is based on maximizing the CP of the FDs, and studies have used the following three different objective functions to assess the coverage level of the FDs:

  - *Demand coverage (covering models)*: The RAFD is based on maximizing the number or percentage of demand covered by the targets in the FD's jurisdiction area [15].
  - *Response time (p-median models)*: Allocation of resources according to the response time. It is a standard defined by FPS authorities and is the time period between receiving the alarm and arriving at the scene. This type of RAFD model tries to minimize it [10].
  - *Population coverage*: The objective function of several RAFD investigations is to maximize the number of people covered under the FD's authority [15].

- *Fire department's performance assessment (FDPA)*: This category consists of the studies that first evaluate the performance of the FDs and then use the results of the performance assessment for RAFD. They conduct the FDPA by assessing the ratio between FPS inputs and outputs, and comparing the FPS outcomes with its targets. For more information on FDPA's methods and variables, please refer to the work by Eslamzadeh et at. (2022) [2].

- *Fire risk assessment (FRA)*: This determines the decision criteria against a predetermined acceptable level of risk by estimating and calculating the fire hazards linked with the occurrence probability and the possibility of the fire and unintended consequences happening [23]. When combined with DC and FDPA, FRA is utilized as an essential variable for RAFD in all 36 reviewed articles; it is also used as the sole RAFD variable in one research.

Considering that some studies used a combination of the above-mentioned approaches, 27 out of 36 reviewed papers used CP (demand coverage 18, response time 5, and population coverage 4), 8 papers used the FDPA approach for RAFD, and only 1 paper applied FRA as its main approach for distributing available resources amongst fire departments.

Although all of the reviewed studies employed incident data in their analysis, the techniques used to analyze the data varied and may be classified into three groups. The groups include those who conducted RAFD using fire incident occurrence data, studies that used incident data to predict future demands and then allocate resources in the FDs' jurisdictions based on the upcoming demand, and finally, those who used sample datasets to evaluate their models and results, due to not having access to the real incident data or for computational reasons. Table 2 shows that 16 studies used historical data without making any predictions for FDPA, 13 studies employed future demand prediction techniques where machine learning algorithms were the dominant prediction method, and 7 studies used sample or computer-generated datasets for their studies. These data reveal that while forecasting future demand is an essential component of the FDs' responsibilities, and decision-makers should develop their strategy to address changing demands in their FPS region, not all of the research investigated this issue.

Since the firefighting operations take place at the station level, 80% of the RAFD papers are conducted at the station level, 14% at the department level (which supervises the stations), and the rest are two papers that are conducted at the state level and the other at the squad level.

### 3.3.2. Equity in RAFD

An important strategic decision of FPS authorities is the usage of the equity method in the RAFD process. In fact, RA decision-makers take equity into account, ensuring that at least a reasonable portion of the resources is distributed evenly across all FDs and eligible areas. This is a critical decision because the fire consequences could be catastrophic [5]. The use of the equity in the RAFD process is described by [5] as the following: "After setting aside a portion of the budget for equity-based allocation, the remaining portion may be distributed based on risk, need, and effectiveness of investment". Despite the importance of considering the equity in RAFD, only 4 out of 36 reviewed RAFD studies incorporated it in their models, and the rest either ignored it or did not use it at all.

### 3.3.3. RAFD Methods

Based on the data in Table 2, except for three papers that used only statistical methods for their analysis, all the rest used a combination of statistical and mathematical methods for FDPA. In total, as depicted in Figure 2, integer/mixed programming (IP), which was used in twenty-one studies, was the most commonly used mathematical RAFD model in the reviewed literature, followed by data envelopment analysis (DEA), which was used in five studies. The main reason for this result is that IP is one of the most common methods for solving CP [15], and DEA is the most frequent method for FDPA [2]. This is the answer to RQ2, which was "*which are the most applied RAFD methods in the studies?*".

Regarding the findings in this section, the FDPA results and the predicted demands in the FDs' jurisdictions are the precondition data for the RAFD models. In addition, due to the risk of catastrophic incidents, applying the equity in the RAFD model to allocate a minimum quantity of resources to all FDs is recommended.

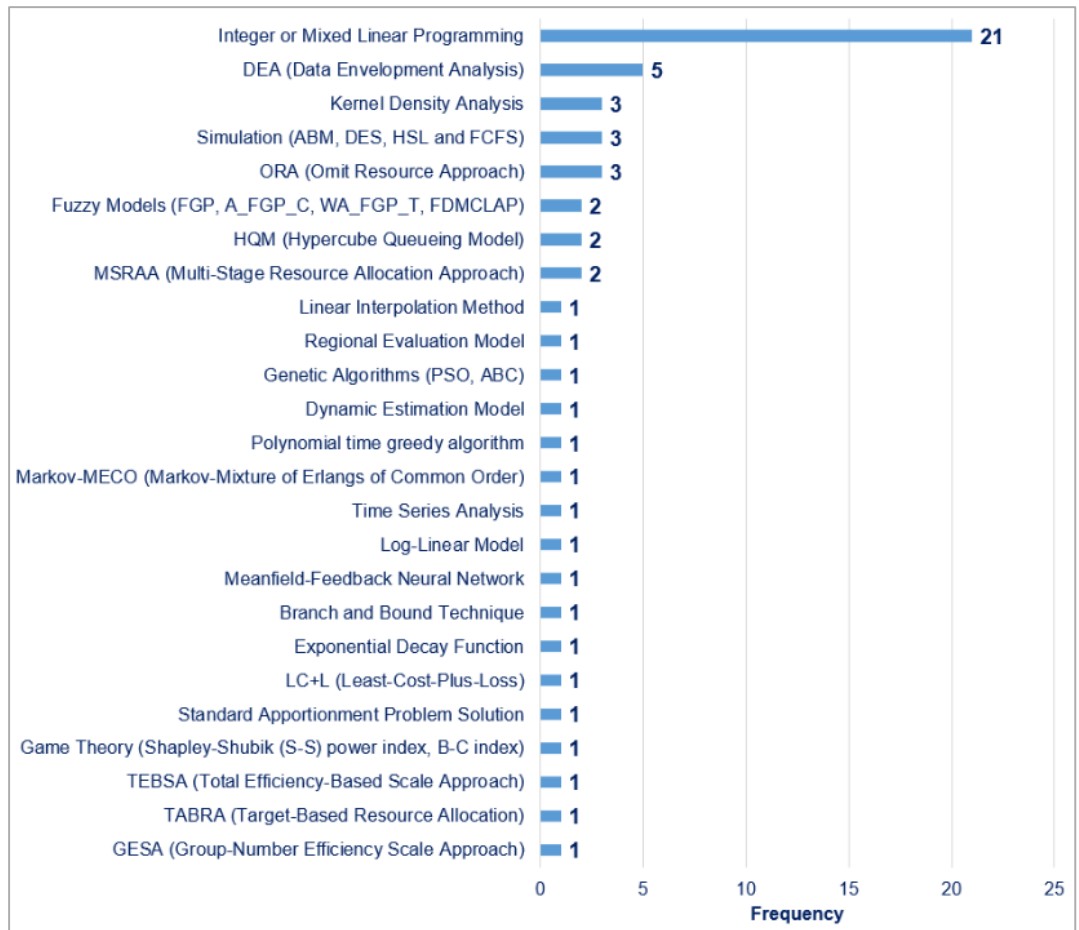

**Figure 2.** The frequency of the mathematical models used for RAFD.

*3.4. Variables*

The findings in this section help to answer the research question 3.

3.4.1. Dependent Variables

The dependent variables in the reviewed papers are shown in Table 2 and varied based on the objective of the papers, some of which had more than one dependent variable. The *FPS's coverage level* was the dependent variable in 21 studies and the RAFD was made based on the calculated values of this variable. The *busy fraction*, which is the proportion of time that the vehicle is busy and not capable of serving a demand point when it is engaged in handling another emergency service call [50,61], was the second most frequent dependent variable that featured in nine studies. As previously mentioned in Section 3, using the performance results of the FDs or FDPA was one of the main approaches for RAFD. Therefore, *FDs' efficiency* was the third most frequent dependent variable that applied to RAFD in seven papers. The *response time* was used as a dependent variable in six articles and was ranked as the fourth one in this section. There were some other dependent variables in five papers, which were only used once, and therefore were not considered as frequent variables in this study (e.g., *effectiveness of nationwide investment* [5] or *fire apparatus requirement score (FARS)* [60]).

3.4.2. Independent Variables

Socioeconomic data (i.e., social and economic characteristics of the investigated area), such as area population, population density, or income rate, and spatiotemporal data, (i.e., spatial and temporal characteristics of fire incidents and FPS) such as date, time, and

location, or fire incidents and station location, are two major categories of independent data that play a very important role in every analysis related to FPS [2,7].

Among the 36 reviewed RAFD articles in this study, 31 papers used either socioeconomic and spatiotemporal variables or both in their RAFD models and analyses. In total, 27 articles used spatiotemporal data of fire occurrences to build the RAFD model, 16 papers included socioeconomic data from jurisdiction regions, and 12 studies combined the two types of data. Even though these variables are influential in RAFD studies, five articles failed to include them in their models.

The reviewed studies employed the 417 independent variables in their RAFD models. Vehicle characteristics, fire incident data, response time data, costs, fire risk indicators, spatiotemporal and socioeconomic factors, and so on were among the independent variables. Figure 3 illustrates the most used independent variables in the coded articles, divided by their criteria and answers to RQ3 (i.e., what are the most employed variables for the RAFD?). As stated in the introductory section, choosing the appropriate variables is a crucial aspect of the RAFD; therefore, the information presented in this section can be beneficial to RAFD assessors and decision-makers.

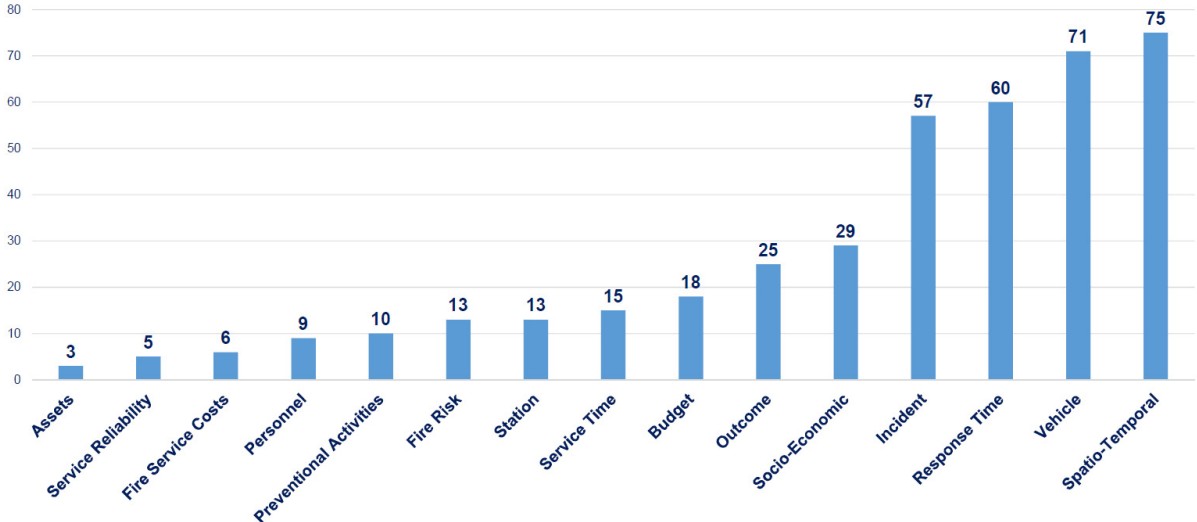

**Figure 3.** The frequency of the independent variables in the assessed FDPA papers.

The most common category of independent variables in the RAFD research, as one might expect, was the spatiotemporal data of fire occurrences, which was utilized 75 times in the articles. The second most frequent category of the independent variable was vehicle specs, which were employed 60 times in the studies, and the third category was response time data, which was applied 60 times in the studies.

From the findings of this section, the most frequent variables were added directly to the general RAFD framework, as they were the main variables for evaluating the RAFD and have been used in most reviewed studies. For instance, population size and population density were the most often used independent variables in RAFD models because of their significant effects on all fire risk indicators, regional demands, FD performance, and the resources needed to meet those demands [1,5,52,54,57]. These significant variables were included in the upper section of the generic RAFD framework in Figure 4.

Section 4 includes a discussion of the independent variables and the suggested general RAFD framework.

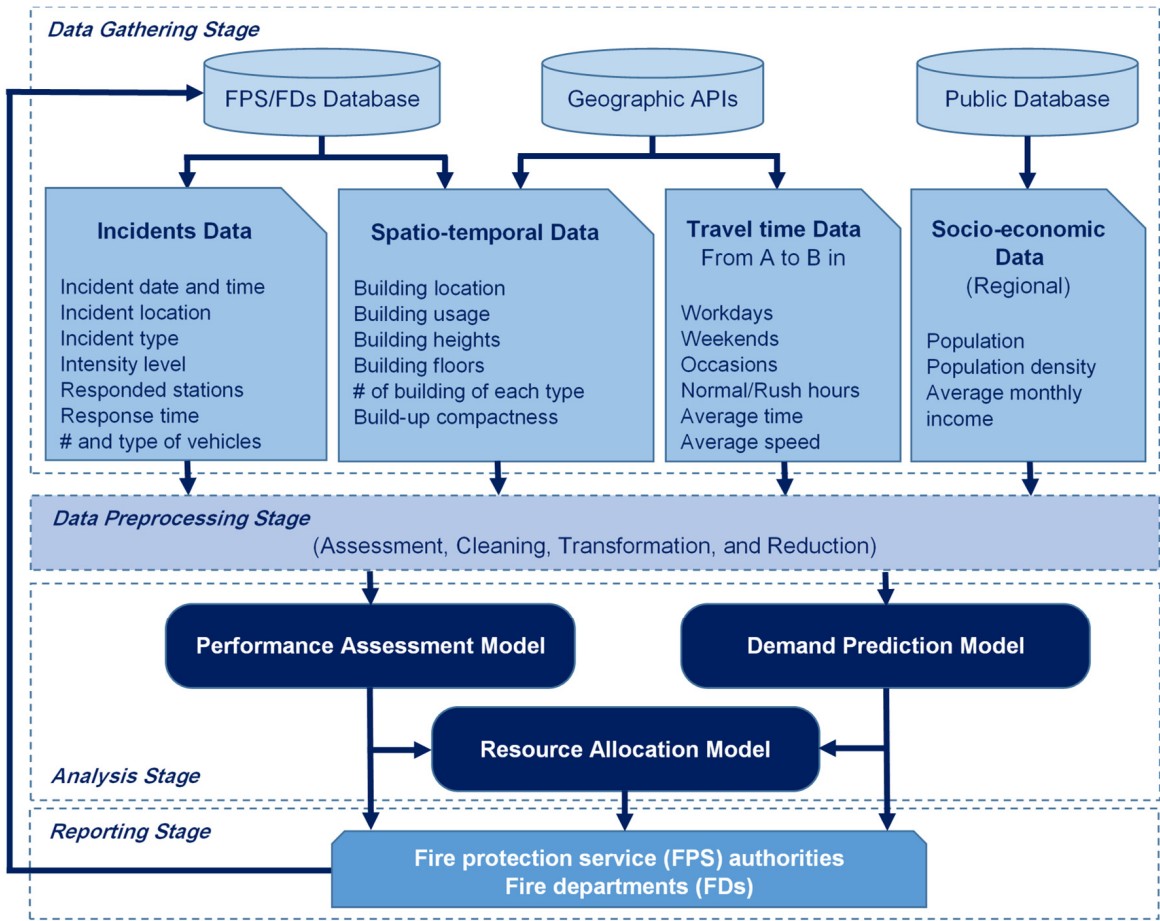

**Figure 4.** General framework for resource allocation in fire departments (RAFDF).

## 4. The General RAFD Framework (RAFDF)

Based on the reported findings of the conducted SLR, a general conceptual framework for RAFD is proposed in Figure 4. Because no distinction was made between RAFD in the public and private sectors in the examined publications, the suggested framework may be applied for both of the FD types.

The suggested general RAFD framework (FRAFD) is structured based on the findings of this study and related studies on fire departments' performance evaluation (FDPA) [2] and effective handling of emergencies in resource constrained urban areas [49].

Data gathering, data processing, analysis, and reporting are the four main stages of the RAFDF. The rest of this section will go through each of these stages in detail.

### 4.1. Data Gathering Stage

Based on the presented results of this research in Section 3, the required data for the RAFD process must be retrieved from public and private databases. For instance, the FPS authorities and FDs (for gathering fire-related data and part of spatiotemporal data), public reports and regional municipalities (for obtaining socioeconomic data), and GIS and online map APIs, such as Google Map Distance Matrix, Open Street Maps, and ArcGIS (for highly vulnerable areas and complementary spatiotemporal data, and calculating travel time), are the most important resources [11,49].

The RAFD assessor and FPS decision-makers may be concerned about the data's availability and accessibility. As a result, with the obtained datasets, the RAFDF may be modified and adapted in practice. The proposed RAFDF does not include the specific indicators in the reviewed paper and only consists of the most frequent variables. Therefore, the RAFD decision-makers are encouraged to consider the distinctive characteristics of their

under-investigation areas and the availability of the relevant data to build their customized RAFD model. For instance, the RAFD decision makers and evaluators should treat HVA locations (such as petrochemical industrial areas, science parks, or industrial zones) as vulnerable areas in their RAFD model because they require more attention and additional FPS backup resources. Section 3.2 (data specifications) provides additional information about HVA.

*4.2. Data Preprocessing Stage*

In this stage, the gathered unconstructed data should be processed, prepared, and translated into a usable format for the analysis stage. The processing stage contains some steps, which are as follows:

- *Data cleansing* is about searching and determining errors in the collected data and then correcting them via different methodologies. The erroneous data could be incomplete, noisy, inconsistent, or unreasonable and data cleansing processes try to improve their quality by finding the missing data, smoothing noise, correcting, or deleting them [62,63].
- *Data integration* combines all data that reside in various sources into one dataset and deals with heterogeneous and redundant data to improve the overall data quality for the analysis stage [62].
- *Data reduction* involves reducing the final dataset size by attribute selection or fitting data into smaller pools, numerosity reduction or using only the data and variables that are relevant to the analysis, and dimensionality reduction or combining similar data [63].
- *Data transformation* involves turning the data into the required formats for the analysis stage and other downstream processes. This step includes aggregating data to a unified format, normalizing the data scales, and smoothing noises [62].

*4.3. Analysis Stage*

After preparing the data in the previous stage, the analysis stage may begin by assessing the performance of the FDs in the performance assessment model [2] and forecasting the fire incidents in the next period (e.g., next month, season, or year) via the demand prediction model [49]. The results of these evaluations will be fed into the resource allocation model, which will provide an RA solution that improves the performance of the FDs while covering the maximum anticipated demands. The results of these two models will be reported to the FPS decision-makers to help them build their plans around the demands of each region and the performance of their FDs. More details are provided in the following sections and are shown in Figure 5.

*4.4. Reporting Stage*

The results of the analysis stage (i.e., the predicted demands data, FDs' performance data, and RA solution) will be communicated with the FPS authorities and organizations so that they may distribute the available resources to the FDs based on their performance and future demands in the respective jurisdictions.

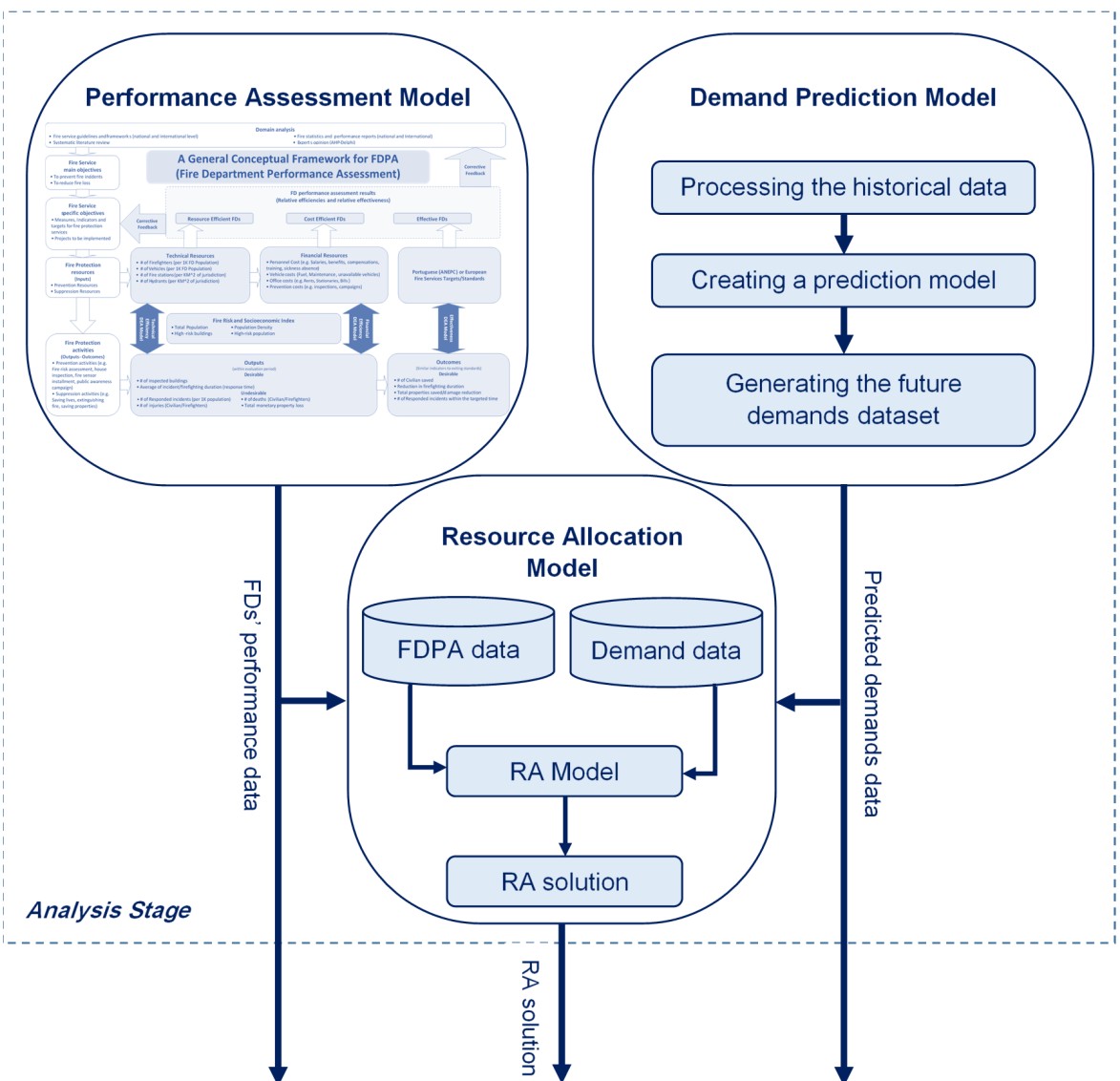

**Figure 5.** The detailed view of the analysis stage of the general RAFD framework (the image of the performance assessment model or FDPA framework obtained from [2]; please read this reference for further details).

## 5. Conclusions

The objective of allocating finite resources among FDs to deliver FPS to more people in less time depends on choosing an appropriate RAFD methodology and variables. Even though several studies have presented various strategies and used a wide range of factors to achieve that aim, not all of them were adequate for RAFD. Hence, a review of RAFD research is necessary to extract RAFD methods and pertinent variables from the literature, determine which are the most common, and propose a general RAFD framework. A five-stage SLR was undertaken in this study to accomplish these objectives, and based on its results, a general framework for RAFD was constructed and presented in detail.

The findings of this study suggest that FDPA (FDs' performance assessment) and forecasting of the FPS demands in the FDs' jurisdiction area are necessary for RAFD assessments. The most prominent analytical techniques are integer programming for RAFD, machine learning for demand prediction, and DEA for FDPA. As part of this study, the RAFD variables were extracted from the evaluated publications and then categorized and arranged according to how frequently they were used in the RAFD literature. The outcomes of this method revealed that the five most common types of RAFD variables

include spatiotemporal data, vehicle specifications, response time, event characteristics, and socioeconomic indicators, respectively.

A general RAFDF was developed by utilizing the obtained knowledge and employing the performance assessment model, demand prediction model, and resource allocation model as its evaluation methods for RAFD. The four main stages of the RAFDF are data collection (to acquire data from published and private resources and experts' opinions), data processing (to clean, integrate, reduce, and transform the unstructured data to a suitable format for the RAFD evaluations), analysis (assessing the FD's performance, forecasting the FPS demands, and finding the best RAFD solution to improve the FDs' performance and cover the maximum demands), and reporting (the results of the analysis stage to the FPS authorities and decision-makers).

The results of this study, coupled with the proposed RAFDF, may shed more light on the RAFD domain and have profound implications on the RAFD process in public and private FDs and FPS by giving the decision-makers access to the most recent developments in this area. The results of this study regarding data collection methods, crucial RAFD evaluation components, frequent RAFD methodologies and indicators, and the suggested framework can help managers and decision-makers of FDs and FPS authorities in both public and private sectors to build their own tailored RAFD model, efficiently allocate their limited resources within their FDs, develop new FPS strategies based on the assessment results, and ultimately raise the level of their service.

Further research might include, but is not limited to, developing FPS strategies at the local and national levels based on the findings of this study and considering the wild and forest fire characteristics and statistics. For interested academics, using and evaluating the RAFDF results in other important disciplines, such as resource management, disaster and emergency management, and RA in the public-private sector might be a new avenue. Furthermore, the RAFDF helps national and local FPS authorities to develop customized RAFD strategies to address their specific geographical needs. As a result, future research may focus on tailoring the RAFDF to regional and national socioeconomic and spatiotemporal factors to foresee and meet the expectations of each region or country.

**Author Contributions:** Literature Review and investigation, M.K.E., conceptualization and methodology, M.K.E., A.G. and P.E.-C.; software M.K.E.; validation, M.K.E., A.G. and P.E.-C.; review and editing, A.G. and P.E.-C. All authors have read and agreed to the published version of the manuscript.

**Funding:** This research was funded by Fundação para a Ciência e a Tecnologia, IP under the grant number [DSAIPA/DS/0088/2019] and the APC was funded by the same grant.

**Institutional Review Board Statement:** Not applicable.

**Informed Consent Statement:** Not applicable.

**Data Availability Statement:** Not applicable.

**Acknowledgments:** The authors would like to thank Fundação para a Ciência e a Tecnologia, IP, for funding this research project (DSAIPA/DS/0088/2019), the ANEPC and its representatives for sharing their knowledge, data and their contribution in validating the results, and the anonymous reviewers for their constructive and valuable suggestions that improved the paper's quality.

**Conflicts of Interest:** The authors declare no conflict of interest.

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
