# Peer review of "A Framework for Resource Allocation in Fire Departments: A Structured Literature Review"

_fire, doi:10.3390/fire5040109_

Round 1

Reviewer 1 Report

This research is to review literature about allocating resources to urban fire departments (FDs) to gain state-of-the-art knowledge of RAFD and identify the most frequent methodologies. The study is very meaningful and the investigation is good. In addition, the structure is clear. However, there are some analysis and comments missing. It is better to add more explanations in each part before the publication.   

1. In SLR Protocol Creation Stage part, the RQ4 (What method is used to aeesess RAFD?) can be added.

2. In 2.3, the investigated studies fall into one of two significant categories: Operational and Strategic. What’s the difference between the operational and strategic? It is a little unclear.

3. Some important information related to the author’s views and analysis is missing. In addition, some summary and comments should be added in each part at result and discussion part..

4. In Figure 10, some words are unclear?

Author Response

Response to Reviewer 1 Comments:

We would like to thank the reviewers very much for reviewing the article and for all the comments and suggestions that contributed to the improvement of the article. We appreciate the hard work you have done and want to acknowledge your tremendous contribution. We hope that we have responded to all comments and made all changes accordingly. Thank you very much.

Below are the answers (in italics) to each comment. In the text, to facilitate the revision of the article, all changes were marked in brown.

We look forward to hearing from you.

Yours faithfully,

The Authors

Reviewer 2 Report

Dear author/s,

The paper is relevant to fire science and addresses a crucial topic in controlling and suppressing wildfires. However, I recommend a profound rewriting of the document by following the following items:

improve the citations: is not homogeneous the references along with the paper. 

Reconsider figures: the plots 2-3-5-6-7 seem irrelevant and could be written as text. 

Figure References: the Tables and Figures should include the references in titles.

Many items: the document is difficult to read because of the inclusion of many items and subitems. Therefore, I recommend dropping the items and developing the writing with more information and content.

More Description: I recommend the authors a better justification of the boolean query.

I encourage the author of the text to modify by following previous comments to achieve a paper publication.

Best regards.

Author Response

Response to Reviewer 1 Comments:

We would like to thank the reviewers very much for reviewing the article and for all the comments and suggestions that contributed to the improvement of the article. We appreciate the hard work you have done and want to acknowledge your tremendous contribution. We hope that we have responded to all comments and made all changes accordingly. Thank you very much.

Below are the answers (in italics) to each comment. In the text, to facilitate the revision of the article, all changes were marked in yellowPlease see the attachment.

We look forward to hearing from you.

Yours faithfully,

The Authors

Reviewer 3 Report

This study reviewed the literature on resource allocation for urban fire departments (FDs) to obtain the latest knowledge on RAFDs and to identify the most common methods and measures studied. Some content need to be revised or additional clarifications before publication.

1. In the section of 2.4. Coding Stage, the professional background of the experts who reviewed the five articles were stated to be required.

2. In the section of 3. Findings and discussion, population density or the size of the population should be an important indicator. These values will be positively correlated with economic, social, cultural and other activities. Usually the fire departments will pay more attention to these areas. Authors are encouraged to do more in-depth research and literature collection on this factor.

3. In the section of 3.2.3 Highly vulnerable areas data, the authors have referred to some literature and explained in-depth analysis. The presentation of the research results is very informative. Some brief description on the content of these literature is encouraged.

4. In the section of 4. The General RAFD Framework (RAFDF), authors are encouraged to indicate whether or not specific factors were found and identified during the analysis. For example, petrochemical industrial areas and science parks in high fire risk areas must have special considerations.

5. In the section of 5. Conclusion, the description of the more concretely results is needed. The content is suggested to be adjusted in this section. The current content still seems to be a general description.

6. In section of 5. Conclusion, please provide suggestions for public and private FDs and FPS managers respectively for better reference.

Author Response

Response to Reviewer 2 Comments:

We would like to thank the reviewers very much for reviewing the article and for all the comments and suggestions that contributed to the improvement of the article. We appreciate the hard work you have done and want to acknowledge your tremendous contribution. We hope that we have responded to all comments and made all changes accordingly. Thank you very much.

Below are the answers (in italics) to each comment. In the text, to facilitate the revision of the article, all changes were marked in Blue. Please see the attachment.

We look forward to hearing from you.

Yours faithfully,

The Authors

Reviewer 4 Report

This is a good application of SLR for  fire department RAFD.

It is not clear which software is used. Please explain.

The last part of the paper for PAM and DPM should be part of another paper. Fig. 9 is not correctly numbered.

Author Response

Response to Reviewer 3 Comments:

We would like to thank the reviewers very much for reviewing the article and for all the comments and suggestions that contributed to the improvement of the article. We appreciate the hard work you have done and want to acknowledge your tremendous contribution. We hope that we have responded to all comments and made all changes accordingly. Thank you very much.

Below are the answers (in italics) to each comment. In the text, to facilitate the revision of the article, all changes were marked in green. Please see the attachment.

We look forward to hearing from you.

Yours faithfully,

The Authors

Round 2

Reviewer 2 Report

Congratulations, you made excellent research and the paper has been improved from the initial document.

Author Response

Dear Reviewer,

We appreciate your evaluation of the paper and all the comments and suggestions that contributed to the article's improvement. We sincerely thank you for your significant contribution and all your hard work. 

Yours faithfully,
The Authors

Reviewer 3 Report

No additional comments.

Author Response

(The authors gave the same response as above.)
